# Metabolic Endotoxemia: From the Gut to Neurodegeneration

**DOI:** 10.3390/ijms25137006

**Published:** 2024-06-26

**Authors:** Mateusz Chmielarz, Beata Sobieszczańska, Kamila Środa-Pomianek

**Affiliations:** 1Department of Microbiology, Wroclaw University of Medicine, Chalubinskiego 4 Street, 50-368 Wroclaw, Poland; 2Department of Biophysics and Neuroscience, Wroclaw University of Medicine, Chalubinskiego 3a, 50-368 Wroclaw, Poland

**Keywords:** metabolic endotoxemia, microglia, neurodegeneration

## Abstract

Metabolic endotoxemia is a severe health problem for residents in developed countries who follow a Western diet, disrupting intestinal microbiota and the whole organism’s homeostasis. Although the effect of endotoxin on the human immune system is well known, its long-term impact on the human body, lasting many months or even years, is unknown. This is due to the difficulty of conducting in vitro and in vivo studies on the prolonged effect of endotoxin on the central nervous system. In this article, based on the available literature, we traced the path of endotoxin from the intestines to the blood through the intestinal epithelium and factors promoting the development of metabolic endotoxemia. The presence of endotoxin in the bloodstream and the inflammation it induces may contribute to lowering the blood–brain barrier, potentially allowing its penetration into the central nervous system; although, the theory is still controversial. Microglia, guarding the central nervous system, are the first line of defense and respond to endotoxin with activation, which may contribute to the development of neurodegenerative diseases. We traced the pro-inflammatory role of endotoxin in neurodegenerative diseases and its impact on the epigenetic regulation of microglial phenotypes.

## 1. Introduction

Metabolic endotoxemia (ME) refers to the condition when bacterial endotoxin, specifically lipopolysaccharides (LPSs), from the gut enter the bloodstream and reach the brain, leading to systemic inflammation. Many recent studies linked ME with neurodegeneration based on elevated endotoxin levels in the blood and brain tissue of patients with neurodegenerative diseases [1]. The role of endotoxin in neuroinflammation is supported by the fact that it is one of the best-known pro-inflammatory bacterial antigens, readily accessible and abundant in human gut microbiota. Additionally, modern research techniques now make it possible to determine in detail the composition and number of bacteria colonizing the human body and the metabolites they produce [2]. Recently, changes in the composition of the intestinal microbiota, termed dysbiosis, have been associated with numerous human diseases, including cardiovascular, kidney, and skin diseases, as well as neurodegeneration. On the other hand, endotoxin often appears in the blood of healthy individuals, e.g., after meals rich in fat, without causing any side effects [3]. Similarly, although associated with numerous diseases, intestinal dysbiosis does not always increase the risk of neuroinflammation. It is also unclear whether endotoxin present in blood can penetrate the blood–brain barrier (BBB) to reach the central nervous system (CNS) (Figure 1) [4]. There are still many unknowns that require a detailed investigation of the role of ME in neurodegeneration. Unfortunately, these studies are hampered by the lack of a good research model for chronic endotoxemia that lasts for months or even years. Despite these challenges, the importance of understanding the role of endotoxin in neurodegeneration cannot be overstated. Our work aims to review the available literature on the impact of endotoxin on neurodegenerative diseases and to trace the subsequent stages of this antigen’s migration from the intestines to the brain. Thus, this article delves into the intricate nature of ME in the context of its potential impact on the CNS, highlighting the most critical factors that contribute to its development. We then delve into how endotoxin infiltrates the bloodstream from the intestines and the intricate immune response it elicits. The article also explores the complicated mechanism of endotoxin transmission across the BBB and the subsequent activation of microglia. Furthermore, we present data on the profound impact of endotoxin-induced microglial activation in prevalent neurodegenerative diseases and the consequential epigenetic changes in microglia. This research is crucial for advancing our understanding of the complex relationship between the gut and the brain, and could lead to new therapeutic strategies for neurodegenerative diseases.

## 2. Metabolic Endotoxemia

ME is characterized by 2-3-fold increased endotoxin levels in the blood and low-grade systemic inflammation without apparent infection [5,6,7]. In obese individuals, ME is usually accompanied by metabolic syndrome, which, in addition to endotoxemia, includes dyslipidemia, hypertension, and insulin resistance altogether, the so-called deadly Norman Kaplan quartet [8]. Metabolic syndrome is a significant cause of cardiovascular disease and has been linked to neurological disorders [9].

The level of endotoxin in the blood of healthy individuals is very low (10 ± 20 pg/mL) [10], but increases in neurodegenerative diseases, suggesting its role in pathological processes in the brain [1,4]. Endotoxemia-induced systemic inflammation and associated cytokines affecting the CNS are the most commonly reported factors influencing depression-like behavior [11]. Numerous experimental studies have shown that humans challenged with endotoxin (0.4–0.8 ng/kg) develop depressive symptoms, such as lowered mood, sadness, irritability, fatigue, anhedonia, and loss of appetite, correlating with the dose of endotoxin [12,13,14,15]. In addition, sleep disturbances, decreased appetite, and deterioration of long-term memory, accompanied by microglia activation and systemic inflammation, confirmed by pro-inflammatory cytokines, including TNFα and IL-6, were observed in individuals challenged with endotoxin [16,17,18]. According to Yirmiya and Goshen [19], peripheral endotoxin application can affect synaptic plasticity and hippocampal-dependent learning and memory. On the other hand, van der Boogaard [3] showed in young, healthy volunteers that, although the administration of endotoxin at two ng/kg resulted in systemic inflammation with high levels of cytokines and increased cortisol levels, it sporadically led to the transient, mild deterioration of brain function without a clinical correlation. Kullmann et al. [20], after the intravenous administration of endotoxin at 0.4 ng/kg to volunteers, also observed an increase in the level of cytokines and a decrease in mood in the subjects. However, the Reading the Mind in the Eyes test performed on the tested volunteers showed additional altered neural activity after endotoxin administration, which, according to the authors, reflected increased responses in the fusiform gyrus, temporoparietal junction, superior temporal gyrus, and precuneus. These increased task-related neural responses after the endotoxin challenge may reflect a compensatory strategy or more excellent social cognitive processing as a function of sickness [20]. After the systemic administration of endotoxin, behavioral changes have also been observed in rodents [21,22]. According to Quin et al. [23], a single intraperitoneal injection of endotoxin caused acute microglial activation in the brains of mice, persisting for at least ten months and resulting in the loss of dopaminergic neurons in the substantia nigra several months later. Zhao et al. [24] showed that the intracerebroventricular injection of endotoxin into mice, even at low doses, induced cognitive impairment through the activation of neuroinflammation. Kurita et al. [25], who investigated the effects of ME after experimental stroke with transient middle cerebral artery occlusion (MCAO) in a murine model of type 2 diabetes, demonstrated that mice presented increased infarct volumes and higher expression levels of endotoxin, TLR4, and inflammatory cytokines in the ischemic brain, as well as more severe neurological impairments and reduced survival rates after MCAO. However, when the animals were administered antibiotics to prevent ME by reducing endotoxin levels, stroke outcomes and reduced neuroinflammation in the ischemic brain were affected.

Many factors, such as poor diet, antibiotics and other medications, lack of physical activity, and environmental factors, may contribute to intestinal dysbiosis and, as a result, to the development of ME [2].

### 2.1. Factors Promoting ME Development

#### 2.1.1. High-Fat Diet

One of the most critical factors contributing to the development of ME is an HFD, the so-called obesogenic diet, poor in unprocessed foods, such as raw vegetables and fruits, grains, and seeds, which are a source of non-fermentable carbohydrates, while abundant in simple sugars and saturated fats. Although the HFD has long been considered an unhealthy diet, associated with numerous metabolic and cardiovascular diseases, it has only recently been observed that endotoxin levels transiently increase after meals rich in fats [26]. Between meals and during periods of starvation, circulating endotoxin levels in healthy individuals drop to undetectable levels, suggesting that fat-laden meals are a source of endotoxemia. Notwithstanding, intervention studies on the effects of an obesogenic diet on endotoxemia in humans present conflicting results. Lyte et al. [27], in healthy, young subjects consuming an HFD containing saturated or unsaturated fatty acids, showed an increase in endotoxin levels, which, however, was not accompanied by an increase in inflammatory markers in the blood, i.e., IL-6 and C-reactive protein. Similarly, Kallio et al. [28] did not observe a relationship between fat intake and fasting endotoxemia levels measured in lean subjects. On the other hand, Herieka et al. [29], analyzing the results of 57 studies on postprandial pro-inflammatory markers and endotoxemia in people on an HFD showed that most studies found endotoxemia and an increase in leukocyte counts in study participants, supporting the hypothesis that an HFD is associated with postprandial pro-inflammatory changes in the blood. Nevertheless, the authors noted that the study groups varied widely in terms of age, BMI, and disease status, which could have affected the analysis results. Ghanim et al. [30], studying a group of healthy subjects aged 20–50 years old, with a BMI < 25, receiving a high-carbohydrate high-fat diet, showed that diet-induced endotoxemia increased free radical levels, NFĸB activation, TLR2, and the expression of the suppressor of cytokine signaling-3 (SOCS-3), in contrast to subjects receiving a high-fiber diet with fruit. According to a study by Pendyal et al. [31], a four-week Western diet increases plasma endotoxin levels by 71%. Likewise, the Western diet in germ-free animals leads to obesity, insulin resistance, and systemic inflammation [28].

#### 2.1.2. Fatty Acids

Subsequently, it has been demonstrated that ME is shaped more by the fat-type consumed than the amount. A systemic review by Candido et al. [32] confirmed that the meal fat profile dictates ME and postprandial endotoxemia fluctuations associated with lipemia. Unsaturated fatty acids, e.g., PUFAs, do not induce post-meal endotoxemia and even reduce it by about 50% in obese individuals. The beneficial effect of unsaturated fatty acids in lowering endotoxemia is associated with the ability of specific fatty acids, e.g., docosahexaenoic acid (DHA) and eicosapentaenoic acid (EPA), to increase chylomicron clearing and reduce serum concentrations of very-low-density lipoproteins (VLDLs), a significant source of triglycerides [33]. In contrast, saturated fatty acids (SFAs), such as stearic and palmitic, increase endotoxin levels in the blood by about 60%. In mice, SFAs potentiate endotoxemia and increase mortality, whereas palmitic acid induces a long-term hyperinflammatory response to endotoxin in SFA-fed animals [34]. In in vitro studies, palmitic acid significantly amplified human aortic endothelial cell inflammatory responses to endotoxin [35]. Similarly, the form of fat consumed and the meal’s composition shape post-meal endotoxemia levels [27,36]. Emulsified fats induce higher levels of endotoxin and stimulate the liver to synthesize bile acids. Fat from the diet is emulsified by bile salts in the gut into micelles, readily incorporating endotoxin. The emulsification of fats enables pancreatic lipase to hydrolyze them into free fatty acids and monoglycerides, which are easily absorbed by enterocytes [32,37]. Moreover, adding healthy foods, such as vegetables, fruits, and nuts, to a fat-rich meal reduces metabolic endotoxemia. Deopurkar et al. [38] evidenced that consuming orange juice with a high-fat high-carbohydrate meal prevented postprandial endotoxemia, likely due to the fiber in orange juice and polyphenol resveratrol, reducing lipemia. Dietary fiber, resistant to digestion by alimentary enzymes, like plant polyphenols and flavonoids, reduces fat and carbohydrate absorption from the gut [39].

On the other hand, according to Mo et al. [40], although it is widely recognized that an HFD affects the increased absorption of endotoxin from the gut into the bloodstream and leads to ME, essentially, the mechanism of systemic inflammation associated with an HFD is unknown and does not necessarily depend on endotoxin. Studies by these investigators suggest that SFAs, through their interactions with TLR4/2 receptors on monocytes and macrophages, can induce systemic inflammation without endotoxin involvement.

#### 2.1.3. Gut Microbiota

Regardless of the involvement of an HFD in endotoxemia, an obesogenic diet certainly impacts the gut microbiota significantly. The disturbed quantitative proportions and depletion of bacterial species diversity observed in intestinal dysbiosis in people on an HFD results in an increase in the release of bacterial metabolites, antigens, and toxins, which exert their toxic effects on the host body over weeks and months, ultimately leading to homeostatic disorders and the development of diseases. Hence, ME-induced obesity appears to result from an imbalance of the intestinal flora. The intestinal microbiota in obese individuals is characterized by reduced diversity and several bacterial taxa. There is a decrease in the abundance of *Bacteroidota* as well as *Bifidobacteria* and *Lactobacillus* beneficial for the maintenance of the intestinal barrier, and an increase in potentially pathogenic bacteria, i.e., *Firmicutes* and *Gammaproteobacteria*, which are the source of endotoxin, leaking from the intestine into the bloodstream [2,41,42,43].

Moreover, according to Zhang et al. [43], obese individuals harbor unique H2-producing bacterial groups, particularly members of the *Prevotellaceae* family that coexist in the gastrointestinal tracts of obese subjects with relatively high numbers of H2-oxidizing methanogenic *Archaea*. The increase in H2-oxidizing methanogenesis increases the conversion of plant polysaccharides to short-chain fatty acids (SCFAs), mainly acetate. In contrast, rapid H2 utilization accelerates polysaccharide fermentation, increasing energy uptake in obese individuals [41]. Additionally, changes in the intestinal microbiota profile activate the echinocannabinoid system, decreasing the intestinal epithelium’s integrity [44]. Fatty acids from the diet also disrupt the intestinal barrier directly by inducing proinflammatory signaling cascades or indirectly via increasing barrier-disrupting cytokines, i.e., TNFα [45]. Nascimento et al. [46], using a mice model, demonstrated that HDF-induced tight junctions (TJs) structure rearrangements (i.e., claudins-1, -2, 3, and ZO-1 decrease) in jejunum epithelia. According to Candido et al. [32], after a single exposure to a high-fat load, which had a subtle impairment in barrier function in non-obese patients, obese subjects demonstrated a more significant increase in the permeability of the small intestine. This suggests SFAs can alter paracellular permeability by directly damaging tight junctions, facilitating the transfer of endotoxin from the intestine into the bloodstream along with chylomicrons, and eventually inducing endotoxemia. An increased abundance of carbohydrate-fermenting *Firmicutes* contributes to the increased biosynthesis of SCFAs, which provide additional energy for the host and are stored as glucose and lipids in adipose tissue. Excessive overeating in obese individuals and the absence of periods of starvation lead to a reduced expression of the fasting-induced adipose factor (FIAF) in adipose tissue [47], a circulating lipoprotein lipase (LPL) inhibitor that increases fat storage in adipose tissue proteins. Another consequence of altering the profile of intestinal microbiota is the activation of the intestinal echinocannabinoid system, mentioned above, and its effect on increasing the permeability of the intestinal epithelial barrier, which facilitates the transfer of endotoxin to the plasma membrane, as well as adipogenesis [48,49,50]. Due to the increased serum concentration of endotoxin in individuals with ME and the demonstrated effect of ingested food on endotoxin influx into the blood, an HFD and existing obesity have the most significant impact on its development.

#### 2.1.4. Obesity

Another important factor determining the effect of diet on ME is body mass index (BMI). The frequent intake of high-carbohydrate and high-fat meals by obese individuals results in impaired FIAF secretion, inhibiting circulating LPL activity. Reduced FIAF secretion induces hypertriglyceridemia, hyperlipidemia, and hypercholesterolemia, increased cellular uptake of fatty acids, and increased tissue lipid storage. FIAF and LPL expression changes affect peripheral and central signals regulating food intake, leading to metabolic syndrome and obesity [41]. According to a study by Backhet et al. [51], the liver contains the independent enzyme AMPK, which controls cellular energy status and activates vital enzymes of mitochondrial fatty acid oxidation. Disturbed gut flora in obese individuals appear to regulate AMPK activity, and its reduced activation results in decreased lipid oxidation and reduced energy expedition.

An HFD, rich in saturated fatty acids and intestinal dysbiosis, lowers the epithelial gut barrier, increasing intestinal permeability and permitting bacterial metabolites, including endotoxin, to leak into the bloodstream [32].

## 3. Endotoxin Transfer from the Gut to Circulation

A single layer of polarized epithelial cells connected by tight junctions provides a highly selective barrier that prevents the passage of harmful contents into the intestinal lumen while allowing the absorption of essential dietary nutrients, electrolytes, and water from the intestinal lumen into circulation [52]. The layer of mucins covering the intestinal epithelium produced by goblet and Paneth cells provides an additional mechanical barrier that prevents luminal bacterial antigens, including endotoxin, from contacting the intestinal epithelium. In addition, intestinal epithelial cells produce alkaline phosphatase (IAP), which, already in the intestinal lumen, inactivates endotoxin by its dephosphorylation [52]. Furthermore, endotoxin in the intestinal lumen stimulates Paneth cells to increase the secretion of antimicrobial peptides (AMPs). In turn, AMP interaction with TLR4 inhibits the inflammatory response [53]. Additionally, a normal human ileal epithelium barely expresses TLR4, minimizing endotoxin recognition; hence, large quantities of luminal endotoxins are usually well tolerated by healthy intestines [54]. Nevertheless, insignificant amounts of endotoxin transiently may appear in the blood of healthy individuals, especially after consuming meals rich in fat, which facilitates the transfer of endotoxin from the intestines into the bloodstream [36,55,56].

Endotoxin crosses the intestinal epithelial barrier via intercellular and transcellular transport.

### 3.1. Transcellular Endotoxin Translocation through the Intestinal Epithelium

Electron microscopy studies indicate that endotoxin as an amphipathic molecule can form micelles that are internalized by enterocytes via clathrin-mediated, lipid raft–caveolae-mediated pathways, via goblet cell-associated passageways (GAPs) or by passive transcellular diffusion [57,58,59,60,61]. Endotoxin taken up by enterocytes enters the cell’s cytosol, bypassing the endosomes [62]. In addition, Gram-negative bacteria often exfoliate outer membrane vesicles (OMVs) containing endotoxin, among other antigens, which are taken up by cells via endocytosis, thus bypassing the lysosomal compartment [62,63]. Taken into the cell’s cytosol, endotoxin is recognized by the NOD-like receptor family, a pyrin domain containing 3 (NLRP3) inflammasomes. Inflammasomes are cytoplasmic, multiprotein complexes composed of sensor proteins and pro-inflammatory caspases that recognize exogenous (PAMPs, pathogen-associated molecular patterns) and endogenous (DAMPs, damage-associated molecular patterns) stimuli. Activation of the inflammasome induces the enzymatic activation of canonical caspase-1, resulting in IL-1β and IL-18 secretion and cell death by pyroptosis in a gasdermin D (GSDMD)-dependent manner, which is an enhancer of pyroptosis [62,64]. Moreover, endotoxin can trigger the activation of non-canonical inflammasome, which results in the activation of caspase-4 and caspase-5. In both cases, however, pyroptosis and IL-1β and IL-18 release occur due to the secondary activation of the canonical inflammasome NLRP3. Additionally, the activation of caspase-4 and caspase-5 produces the high-mobility group protein 1 (HMGB1) and IL-1α [65]. Thus, non-canonical inflammasome activation primarily drives endotoxin-induced cell death, whereas canonical inflammasome activation leads to an inflammatory response [66,67,68].

The transcellular pathway, and primarily the GAPs pathway, represents the main pathway for endotoxin transport to dendritic cells in the basal lamina, where, through a TLR4-dependent pathway, endotoxin can increase the permeability of the intestinal epithelium, enabling the paracellular transport of endotoxin into the blood [57].

Another route of transcellular transport of endotoxin from the intestinal lumen to the blood occurs through chylomicrons. The intestinal epithelial cells internalize endotoxin from the apical surface and transport it to the Golgi apparatus, where newly formed chylomicrons are located [69]. The high affinity of endotoxin for chylomicrons ensures its internalization and basolateral secretion along with the chylomicrons. Hence, most of the endotoxin absorbed from the gut is present in the chylomicron remnants. Thus, endotoxin can cross the small intestinal epithelium under physiological conditions without inducing changes in the permeability of the intestinal epithelial barrier [58]. In addition, the formation of chylomicrons promotes endotoxin transport through mesenteric lymph nodes [61,69]. This type of endotoxin transport can occur under physiological conditions in healthy individuals after consuming fat-rich meals (Figure 2).

### 3.2. Paracellular Endotoxin Transport through the Intestinal Epithelium

The intercellular translocation of endotoxin occurs upon reduced integrity of the intestinal epithelium maintained by desmosomes, adherens junctions (AJs), and TJs located at the peak lateral junctions of membranes of neighboring cells and along the lateral membranes [52]. Paracellular transport occurs either through the pore pathway or through the leak pathway. The pore pathway serves to transport ions, water, and small particles, regardless of their charge. It is regulated by the claudin family of proteins and selective transporters of amino acids, electrolytes, and sugars [61,70,71]. The leak pathway transports larger particles, regardless of their cargo, and is regulated by the pore pathway. Leak pathway transport is influenced by inflammation-related cytokines, such as TNFα and IL-13, which affect tight junctions [61]. Nighot et al. [72] on Caco-2 cells and C57BL/6 mice with TLR-4 and MyD88 knockout showed that endotoxin at clinically achievable concentrations (ranging from 0 to 2000 pg/mL) induced an increase in intestinal epithelial paracellular permeability, which was preceded by the activation of transforming growth factor-β-activating kinase-1 (TAK-1) and canonical NFκB (p50/p65) pathways. The effect was inhibited by a pharmacological inhibitor of TAK-1, oxozeaenol, confirming that TAK-1 activation was required for the endotoxin-induced increase in intestinal permeability. This indicates that endotoxin, even at low concentrations, can increase intestinal epithelial permeability through an increase in the expression of TJ effector protein myosin-light chain kinase (MLCK), which induces the opening of the TJ barrier by activating myosin light chain (MLC) phosphorylation, and the Mg+2/myosin ATPase-dependent contraction of perijunctional actin and myosin filaments, which in turn generates mechanical tension and centripetal pulling apart of the TJ barrier [72].

However, the presence of endotoxin in the blood does not necessarily lead to any clinical consequences due to the body’s defense mechanisms that rapidly remove it from circulation [70,71].

### 3.3. Endotoxin Neutralization in the blood

Endotoxin entering the portal vein from the intestines is rapidly removed by liver hepatocytes with bile salts back to the intestines. The endotoxin binding to chylomicrons accelerates its hepatic uptake and removal from circulation. Both chylomicrons and endotoxin-binding bile salts inhibit the pyrogenic activity of endotoxin [58,73]. In addition to endotoxin detoxification in the liver, during circulation, endotoxin is efficiently bound by numerous proteins, i.e., LBP, sCD14, plasma lipoproteins, adiponectin, serum amyloid P, lectins, gelsolin, and hemoglobin [57,74]. LBP, a member of the acute-phase proteins, is essential for removing endotoxin from circulation, with levels rising during the acute phase of inflammation. LBP facilitates the transfer of endotoxin into immune cells and its degradation and catalyzes the transfer of endotoxin to chylomicrons and high-density lipoprotein (HDL), which promotes its detoxification in the liver [57]. The cationic plasma proteins, bactericidal permeability-increasing protein (BPI), and lactoferrin, secreted by neutrophils during inflammation, inhibit endotoxin endotoxic activity [75]. Antimicrobial peptides (AMPs), produced at the inflammation site and essential to innate immune mechanisms, serve a similar function [76,77,78]. AMPs rich in cationic and hydrophobic amino acids bind and neutralize the negative charge of the endotoxin molecule. In addition, AMPs interfere with the TLR4 recognition system, which inhibits cytokine production in response to endotoxin. In addition, AMPs modulate immune cells’ response to endotoxin and reduce endotoxin-induced NO and TNFα levels, preventing tissue damage [79,80]. Among the endotoxin-binding plasma lipoproteins, i.e., LDLs, HDLs, VDLs, triglycerides, and chylomicrons, which reduce its biological activity, LDLs appears to play the most crucial role, neutralizing the endotoxin [81]. Endotoxin bound to plasma lipoproteins does not bind to macrophages and is 100- to 1000-times less active in monocyte activation than free endotoxin molecules [82].

Despite the numerous endotoxin neutralization mechanisms, transient high endotoxin levels in the blood during acute infections, similar to low concentrations accompanying ME, can induce a cascade of immune reactions.

## 4. Endotoxin Structure Dictates the Host Immune Response

Endotoxin, as a component of the outer membrane of Gram-negative bacteria, is released into the intestinal lumen during bacterial growth or breakdown. Although the microflora of the mucous membranes of the mouth, respiratory tract, and skin include Gram-negative bacteria, the endotoxin released by the intestinal microflora, which contain about 50% Gram-negative bacteria in its composition, is commonly considered the source of ME [1,6,36,58]. Intestinal flora can produce 2–50 mg of endotoxin daily. Humans are especially sensitive to endotoxin, with the lethal intravenous dose being as low as 1 to 2 μg [83]. On the other hand, levels of endotoxin up to 5 pg/mL are detected in the plasma of healthy individuals without causing any clinical symptoms [84]. Endotoxin entering the circulation in high concentrations (>10 ng/mL) induces an acute resolving immune response. In contrast, low concentrations of endotoxin (1–100 pg/mL) cause chronic, non-resolving inflammation, contributing to chronic heart disease, diabetes, rheumatoid arthritis, and the development of neurodegenerative and psychiatric disorders, among others [85]. Although high doses of endotoxin induce the synthesis and secretion of pro-inflammatory cytokines, they simultaneously stimulate the pendulous secretion of anti-inflammatory cytokines, leading to the resolution of inflammation and developing a tolerance to endotoxin. Small doses of endotoxin have the opposite effect, as they primarily potentiate the pro-inflammatory response, referred to as an endotoxin priming impact, i.e., they stimulate immune cells to produce higher levels of proinflammatory cytokines than higher doses of endotoxin. Mice treated with lower amounts of endotoxin show higher mortality than mice administered higher doses. However, this phenomenon’s mechanisms are poorly understood [86].

Endotoxin is an amphipathic glycolipid in the outer membrane of Gram-negative bacteria, composed of toxic lipid A and a carbohydrate region. Hydrophobic lipid A, which anchors the endotoxin molecule to the bacterial outer membrane, consists of a phosphorylated glucosamine backbone with acyl chains attached by an ester or amide linkage. In the carbohydrate region of endotoxin, two areas are distinguished: the core oligosaccharide and the O antigen. The core oligosaccharide is relatively conserved among different bacterial species. The O antigen consists of multiple copies of repeated carbohydrate units with other structures [86,87]. The toxic and immunogenic activity of endotoxin is determined by the number of acyl chains in lipid A, which varies by species and even bacterial strain [88].

Cells in constant contact with endotoxin produce a state of tolerance that results in specific immunosuppression protecting the host from cytokine-induced tissue damage and systemic response; hence, the repeated exposure of cells to endotoxin also abolishes its pyrogenic effect [84]. In the interaction of the most common hexaacylated lipid A molecules of endotoxin with TLR4, the first five acyl chains are buried in the hydrophobic cavity TLR-4-adaptor molecule MD-2, while the sixth chain dopes the binding to TLR-4 (the precognition for TLR4-mediated intercellular signaling) [87,89,90]. In contrast, endotoxin molecules in which lipid A contains five or four acyl chains have a drastically reduced ability to activate TLR-4-mediated signaling and, thus, the ability to induce an inflammatory response. Penta- or hepta-acyl ligands are 100-fold less active, while tetra-acyl analogs are inactive [91]. On this basis, endotoxin molecules possessing hexaacylated lipid A and inducing a robust inflammatory response are referred to as TLR4 agonists. In contrast, endotoxin molecules containing 5 or 4 acyl chains are referred to as weak TLR-4 agonists or antagonists. Lipid A, lacking the sixth acyl chain, has inadequate or no pro-inflammatory effects. Hence, endotoxin molecules that are TRL-4 antagonists, while blocking but not activating the TLR-4 signaling cascade, may exert beneficial effects by inhibiting the immune response by blocking this receptor for hexaacylated endotoxin molecules [90,92]. The ratio of agonistic (toxic) and antagonistic (highly toxic) endotoxin molecules in the gut is thought to play a critical role in regulating homeostasis. The prevalence of Gram-negative species of bacteria presenting highly toxic endotoxin in the intestinal microbiome can promote the TH1/TH17 response and induce inflammation, lowering the intestinal barrier and leakage of pro-inflammatory endotoxin into the bloodstream [90]. In turn, the postprandial leakage of a mixture of high- and low-toxic endotoxin particles from the gut can lead to blocking the inflammatory response induced by the high-toxic endotoxin by binding to TLR-4 of the low-toxic endotoxin. As a result, the type of lipid A endotoxin molecule entering the blood will dictate the consequences of metabolic endotoxemia, i.e., the lack of immune response or inflammation [93].

In addition to the acyl chains, the number of phosphate groups also influences the immunogenic properties of lipid A. Most lipid A molecules have two phosphate groups bound to sugar residues coupled to glucosamines. A loss of one or both groups inactivates the endotoxic activity of endotoxin a hundredfold. Dephospho-endotoxin still binds to TLR4, but acts as an antagonist in this form. Thus, reducing endotoxin toxicity through lipid A dephosphorylation inhibits intracellular signaling, NFĸΒ activation, and secretion of pro-inflammatory markers [52]. Nevertheless, some pathogens deliberately do not produce phosphate residues in lipid A, hence avoiding an immune response [90].

### Endotoxin as a Potent Inductor of Proinflammatory Response

Endotoxin entering circulation, if not neutralized, induces inflammation, even in small doses. In the circulation, the first cells recognizing endotoxin as PAMP molecules are phagocytic cells, i.e., monocytes/macrophages, neutrophils, and dendritic cells, presenting a pattern recognition receptor (PRR) family, e.g., TLR4, NOD-like receptors (NLRs), RAGE, and TREM1 [86,94]. Endotoxin is also recognized by receptors presented on macrophages, such as CD14, macrophage scavenger receptors (SRs), and β2 leukocyte integrins (CD11a/CD18, CD11b/CD18, and CD11c/CD18) [1,95,96]. TRL4 and CD11/CD18 receptors can initiate a pro-inflammatory response cascade in macrophages. In contrast, SRs typically act as accessory proteins to facilitate endotoxin binding and inactivation. SR-A1 receptor binds endotoxin, and lipid A facilitates the internalization of endotoxin through the lipid–raft pathway and, through interaction with TLR4 and NFĸΒ activation, participates in the adaptive immune response. The SR-B1 receptor presented by liver cells facilitates endotoxin binding from the blood and its neutralization. In contrast, the SR-J/RAGE (receptor for advanced glycation end-product) surface receptor binds outside the HMGB1 protein, which, as a RAGE ligand, participates in the endotoxin-induced activation of the inflammatory response in macrophages. RAGEs are primarily involved in chronic inflammation [96,97].

Endotoxin, a potent agonist, plays a pivotal role in activating the TLR4 receptor. This receptor, the only TLR receptor that activates two distinct, i.e., MyD88-dependent and TRIF-dependent, signaling pathways, is triggered via the recruitment of adaptor proteins, i.e., the acute phase endotoxin-binding protein (LBP) and glycosylphosphatidylinositol-anchored membrane protein (CD14, a cluster of differentiation-14) that transfers endotoxin into hydrophobic pockets within the MD2 (myeloid differentiation 2) bound to TLR4. CD14 is expressed on the surface of macrophages and leukocytes. In CD14-negative cells, e.g., endothelial cells, soluble CD14 present in serum functionally replaces membrane-bound CD14 [98]. The TLR4/MD2 complex is activated upon endotoxin binding, initiating a signal that leads to TLR4 homodimerization via the toll–interleukin-1 receptor (TIR) domain-containing adaptor protein TIRAP. This process results in conformational changes in the TLR4 molecule. In the MyD88-dependent signaling pathway, the signal transducer MyD88 (myeloid differentiation factor 88) is recruited through the TIRAP (also known as Mal, MyD88 adaptor-like). The recruitment of MyD88 leads to its interaction with a complex of IL-1R receptor-related kinases, i.e., IRAK1, IRAK2, and IRAK4, which bind to MyD88 via homophilic death domains (DDs) (Figure 3). The MyD88/IRAKs complex recruits E3 ubiquitin ligase tumor necrosis factor receptor-associated factor 6 (TRAF6) to form the Myddosome complex. The phosphorylated IRAK/TRAF6 complex dissociates from the receptor and forms a new complex with transforming growth factor β (TGF-β)-activated kinase (TAK1) bound to TAB-1 and TAB2 proteins, which phosphorylate TAK1. TAK1, in turn, phosphorylates the inhibitor of NFĸΒ, IkΒ kinase complex (IKK), and mitogen-activated kinase (MAPK) kinases 3 and 9 (MKK3 and MKK9). IkΒ phosphorylation allows nuclear NFĸB transcription factors (p50 and p65) to translocate to the nucleus and bind to promoter regions to express inflammatory cytokines and chemokines, i.e., TNFα, IL-1, IL-6, IL-8, IL-12, and COX-2 [99,100,101,102,103,104,105]. In turn, MKK3 and MKK9 phosphorylate p38; ERK, a c-jun N-terminal kinase (JNK) MAPK; and activate activator protein-1 (AP-1) transcription factors, which control several cellular processes, such as differentiation, proliferation, and apoptosis, and activate inflammatory gene expression in the nucleus [99,106].

The MyD88-independent signaling pathway, also known as the TRIF-dependent pathway, involves the TRIF domain (TIR domain-containing adapter inducing IFNβ) and TRIF-related adapter molecule TRAM, bridging TRIF to TLR4. The initiation of the TRIF-dependent signaling pathway requires endotoxin-induced TLR4 endocytosis, which CD14 supports [107]. TLR4 internalization is initiated by decreased phosphatidylinositol 4,5-biphosphate PI(4,5)P2 levels, which induces TIRAP translocation to the cell membrane. The TRIF-dependent signaling pathway is initiated by TRIF interacting with TRAF6, which recruits the kinase-receptor-interacting protein 1 (RIP-1), which in turn interacts with the TAK1 complex, activating NFĸB and MAPKs and the induction of inflammatory cytokines. Moreover, TRIF promotes the TRAF3-dependent activation of the IKK-related kinase TANK-binding kinase 1 (TBK1) and IKKi, along with NEMO (IκB kinase γ) for the phosphorylation and dimerization of the IFN-inducing transcription factor IFN regulatory factor 3 (IRF3). The IRF3 homodimer translocates to the nucleus to regulate the expression of type I IFN genes and other inflammatory mediators, including RANTES and CXCL10, an IFNγ-inducible protein 10 [98,108,109,110,111]. The MyD88-independent pathway also activates the production and secretion of TNFα, leading to late-phase NFκB activation through IRF3 and TNFα secretion [107]. Ultimately, the endosome is lysosomaly degraded, which terminates the inflammatory response. Therefore, MyD88-independent signaling is essential for the induction of adaptative immune response and accounts for most of the endotoxin responses [101,112]. The MYD88-dependent and TRIF-dependent (MYD88-independent) signaling pathways are triggered consecutively [52,101,102,103,112].

The ability of endotoxin to activate a number of signaling pathways, which leads to inflammation, just as the presence of endotoxin in the blood, does not necessarily enable its transfer to the CNS, guarded by tight structural and immunological barriers in healthy conditions [113].

## 5. Can Endotoxin Cross the Blood–Brain Barrier?

Endothelial cells forming the BBB are structurally and functionally different from endothelial cells (ECs) lining peripheral blood vessels. First and foremost, the endothelium in the brain blood vessels exhibits much tighter junctions formed by TJs and AJs, which prevent the passage of polar particles through the BBB (Figure 2). Moreover, ECs forming the BBB lack transcellular channels and fenestrations in contrast with peripheral ECs [113,114,115,116,117,118]. Additionally, ECs in the BBB are characterized by low and slow pinocytic activity, limiting transcellular passage, which is further tightly regulated by active transport systems, making the barrier extremely selective [119]. The BBB structure is further reinforced by basal lamina formed predominantly of collagen type IV, laminin, and heparan sulfate proteoglycan, containing metalloproteinases regulating BBB function [120,121]. The basement membrane of the endothelial capillaries is encircled by protrusions of glial cells, astrocytes, and phagocytosis-capable pericytes, ensuring the precise regulation of the composition of the environment surrounding neurons. This specific BBB structure prevents the passage of harmful, toxic substances into the brain’s neural tissue [64,106,120,122]. However, the BBB changes around the circumventricular organs (CVOs) located adjacent to the third and fourth ventricles of the brain. CVOs present fenestrated capillaries with loosely connected astrocytic protrusions, allowing molecules to travel freely between the blood and the CVO tissue and making neurons more susceptible to peripheral signals. Nevertheless, CVOs are protected by tanycytes presenting tight junction proteins, forming a barrier guarding the ventricular organs, and controlling the diffusion of blood-borne substances to the CSF [123].

In addition to being a selective neuroprotective physical barrier, the BBB plays a vital role in the CNS immune response to internal and external factors. The endothelial cells of the BBB produce numerous cytokines, chemokines, and growth factors, whose role is to recruit lymphocytes and monocytes and regulate the local immune response [124,125]. The BBB cells also present intracellular and vascular cell adhesion molecules involved in the adhesion and transmigration of leukocytes to the CNS (Figure 4) [114,118,124]. Hence, the neuroprotective functions of the BBB can be impaired by systemic inflammation induced by infection or toxic metabolites [119]. The BBB responds to systemic inflammation in several ways, e.g., changes in signaling, enhanced cellular traffic, an increase in solute permeability, and direct damage [116,125].

The endotoxin molecule, classified as a large oligosaccharide polymer with negatively charged and hydrophobic lipid A, cannot cross the intact BBB [4]. In addition, the low expression of TLR4 receptors in microglia and astrocytes provides specific protection for the brain against endotoxin-induced inflammation. Nonetheless, several animal studies demonstrated that radioactively labeled endotoxin administered intravenously to mice crossed the BBB by binding reversibly to brain endothelial cells [126]. Repeated injections of endotoxin2 did not increase endotoxin uptake or its passage into the brain, suggesting it is unlikely that a dose of endotoxin sufficient to induce neurodegeneration would pass into the brain [127]. Similarly, the intraperitoneal administration of low doses of endotoxin did not compromise the BBB nor induce an innate immune response in the model of a neonatal rat brain [128]. Opposite animal studies have shown that endotoxin infiltrates the brain under physiological conditions via a lipoprotein-mediated transport mechanism and is detected in brain structures, such as CVOs, choroid plexus, meningeal cells, astrocytes, tanacytes, and ECs.

Endotoxin is also detected postmortem in the brains of humans suffering from neurodegenerative diseases, suggesting that the molecule, in particular circumstances, can penetrate the BBB [129,130]. Zhan et al. [131] demonstrated elevated levels of *E. coli* endotoxin in the brain parenchyma and vessels in patients with AD compared to control brains. Endotoxin presence was also confirmed in the neocortex, hippocampus, and superior temporal lobe of AD-affected brains [132,133]. Multiple mechanisms of endotoxin penetration across the BBB are considered, and many of these routes have been confirmed in in vitro studies. First, endotoxin in the bloodstream is bound by the acute phase protein LBP, which may allow the passage of endotoxin to the CNS through receptors present on ECs of the BBB, such as TLR4 receptors, scavenger receptor class B type 1 (SR-B1), apolipoprotein E receptor 2 (ApoER2), or low-density lipoprotein receptor (LDLr) [1,129,134,135]. Second, endotoxin-induced systemic inflammation stimulates the secretion of pro-inflammatory cytokines and the expression of adhesion molecules on BBB endothelial cells, allowing the infiltration of the barrier by endotoxin-carrying peripheral immune cells [129,136,137]. Third, endotoxin can cross the BBB within outer membrane vesicles (OMVs), exfoliated from the surface of Gram-negative bacteria, which act as messengers, carrying bacterial antigens to the external environment and host cells [129,138]. Nonaka et al. [139] showed that *Porphyromonas gingivalis* OMVs are internalized by the human cerebral endothelial cell line (hCMEC/D3) and destroy TJs, ZO-2, and occludin proteins. Similar findings were reported by Ha et al. [140], who confirmed the transport of *Aggregatibacter actinomycetemcomitans* OMVs across the BBB. Likewise, endotoxin-containing OMVs can be taken up by ECs of the BBB, activating inflammasomes and inducing their pyroptosis, thus opening the route for endotoxin [141]. Although the load of OMVs varies and depends on the species of the microorganism, endotoxin is the most common virulence factor carried by the OMVs of Gram-negative bacteria [142,143].

These data indicate that endotoxin can penetrate the CNS, but only in the case of a disrupted BBB. But can ME and low blood endotoxin levels contribute to BBB leakage?

### The Impact of Endotoxin on the Blood–Brain Barrier

The ECs of the BBB interface between the blood and CNS are exposed to circulating endotoxin and cytokines during ME. Peripheral cytokines can directly disturb the BBB, impacting ECs and activating them to secrete proinflammatory chemokines, i.e., monocyte chemoattractant protein-1 (MCP-1), interferon gamma-induced protein 10 (IP-10), and adhesion molecules, i.e., VCAM-1 and ICAM-1, recruiting leukocytes and increasing the transcellular permeability of the BBB [117]. Attracted leukocytes encountering the ECs send signals directing the junctional rearrangement to loosen the endothelial barrier. These signals trigger increased intracellular calcium levels, causing the activation of myosin light chain kinase (MLCK), which leads to endothelial cell retraction, facilitating leukocyte passage [144].

The best-studied cytokines produced upon endotoxemia and affecting BBB permeability include TNFα and IL-1β. Both these cytokines induce long-term changes in human cerebral microvascular endothelial cells (hCMECs), such as the increased cell-surface expression of key leukocyte-adhesion molecules, i.e., CD62E, VCAM-1, and ICAM-1, secretion of a panel of chemokines, growth factors, soluble adhesion molecules, and receptors, contributing to reduced BBB in a concentration and time-dependent manner. Interestingly, the barrier resistance increased over days, indicating that the response of hCMECs to both these cytokines strengthened their integrity [145,146]. In an animal model, Browyer et al. [147] demonstrated that subcutaneous endotoxin injection activated the microglia in the hippocampus and cortex, but with no pathological inflammation. Still, proximal elongated processes of microglial cells were closely associated with brain vasculature, suggesting that vascular damage triggers microglia migration to repair the affected vessels. On the other hand, according to the authors, diverting microglial interactions away from synaptic remodeling and other microglial interactions with neurons may adversely affect neuronal function. This is a crucial insight that could guide future research in this area. Moreover, whether the long-term, multi-month effects of ME and endotoxin on the BBB similarly impact the inflammatory milieu in microglia is unknown.

Endotoxin circulating during ME can also disrupt the BBB through the TLR4 receptor on BBB endothelial cells [1]. TLR4 activation in hCMECs leads to pro-inflammatory mediators’ synthesis and secretion, disrupting TJs and increasing BBB permeability [146,148,149,150]. Qin et al. [151] evidenced that endotoxin, through TLR4 activation, elevates phosphorylation of p38MAPK and JNK kinases and decreases occludin mRNA levels while enhancing metalloproteinase-2 (MMP-2) expression. MMP endopeptidases cleave most extracellular matrix components, including fibronectin, laminin, proteoglycans, and type IV collagen, influencing the regulation of TJ protein expression [152,153,154,155,156,157,158].

Furthermore, endotoxin can disrupt the BBB by stimulating ROS production in ECs lining the barrier. Oxidative stress, by increasing intracellular Ca+2 levels, activates MAP kinases, which are responsible for the phosphorylation of TJs and their redistribution [159,160].

These findings, which underscore the intricate relationship between the BBB and microglial response, are profoundly significant in advancing our understanding of neurological conditions. Hence, it is essential to understand the effects of ME and endotoxin on microglia.

## 6. Metabolic Endotoxemia-Related Microglia Activation

Even though ME is characterized by a low level of endotoxin in the blood, its concentration may increase significantly after meals rich in SFAs via endotoxin-loaden chylomicrons [40]. Hence, the endotoxin level in ME is subject to significant fluctuations from low to very high levels. Moreover, in ME, variable endotoxin levels are continuously present in the blood for many months or years, making it difficult to study its effects on microglial molecular patterns in vitro and in vivo on animal models. Nevertheless, in the studies of neurodegenerative diseases, endotoxin, as a common readily available microbial antigen, remains considered the primary cause inducing the inflammatory reaction underlying the pathogenesis of these diseases [11]. Two main lines of research dominate the study of the role of endotoxemia in the development and progression of neurodegenerative diseases.

The first line of studies is related to the theory of physiological inflammation or immune tolerance, which assumes that a series of physiological immune reactions lead to the maintenance of homeostasis and recovery from pathological inflammation [161,162]. According to this idea, microglia preconditioning with low-dose endotoxin induces beneficial immune responses rather than pathological inflammation, thus contributing to neuroprotection. Several studies were undertaken to reveal the neuroprotective mechanism of endotoxin by conditioning microglia with low doses of endotoxin to induce their anti-inflammatory transformation and tolerance phenotype microglia [161,163,164,165].

Reiterating the key findings, the second line of studies demonstrates that priming microglia with repeated, even mild, stimuli can promote trained microglia and an enhanced inflammatory response, playing a crucial role in the progressive reduction in white matter, autoimmunity, and cognitive decline [166]. These findings provide a comprehensive understanding of the microglial immune response to endotoxin, which is dose-dependent. At high doses, endotoxin triggers a ‘cytokine storm’, a potentially deadly systemic inflammatory response syndrome (SIRS). Lower doses (in the 1–100 ng/mL range) can induce a state of tolerance to subsequent toxic doses of endotoxin. However, extremely low doses (in the 0.05–0.5 ng/mL range) have the opposite effect, priming the immune system for an even more violent response to subsequent challenges [167]. On microglial cells derived from a neonatal mouse brain, Lajqi et al. [168] demonstrated that prolonged priming microglia with ultra-low but increasing endotoxin doses (from 1 fg/mL to 100 ng/mL) provoked trained immunity with an increased production of pro-inflammatory markers. The opposite reaction, i.e., microglia tolerance, was observed when microglial cells were treated with higher and rising doses of endotoxin (from 1 fg/mL to 1 µg/mL) for 24 h. The endotoxin-induced inflammatory response of microglia can also be responsible for the delayed but progressive loss of dopaminergic neurons in the SN of the brain. Applying a conditioned medium from endotoxin-activated microglia cultures to primary hippocampal neurons has been shown to induce the loss of synapses [169,170].

Similarly, peripheral inflammation can lead to an enhanced microglia response or inhibition of its activity, exacerbating or alleviating pathology in the mouse brain [163]. In an adult mouse model, endotoxin-mediated peripheral inflammation has been shown to cause long-term changes in the microglia response, despite the apparent return to homeostasis between stimuli [163].

## 7. Role of Endotoxin in Neurodegenerative Diseases

The incidence of incurable neurodegenerative diseases (NDDs), such as AD, Parkinson’s disease (PD), amyotrophic lateral sclerosis (ALS), frontotemporal dementia (FTD), and Huntington’s disease (HD), is constantly increasing. It currently affects approximately 15% of the world’s population [171,172]. The likelihood of developing neurodegenerative diseases increases with age. Hence, the aging population is considered the main factor causing the increase in the incidence of these diseases. Although the pathogenesis of most NDDs has been more or less elucidated, the pathomechanisms of these diseases remain to be fully revealed. It is widely accepted that the pathogenesis of neurodegeneration is multifactorial and includes inflammation, genetic, and epigenetic factors [171,172,173,174,175]. Even though each NDD has a different clinical course and unique disease-related markers, the common feature is chronic neuroinflammation affecting distinct specific neurons in particular CNS regions [173,176].

The hypothesis of the role of endotoxin in neurodegeneration is mainly supported by increased endotoxin concentrations in the blood of patients with PD, AD, ALS, and HD [1]. Numerous in vitro studies on primary neuronal cell lines and in vivo studies on rodents confirmed that endotoxin is an efficient stimulant of neuroinflammation via microglial activation within the CNS [1]. It has been discovered to trigger an increase in Aβ production, accumulation, and hyperphosphorylation of tau protein and α-synuclein [1,177,178,179].

### 7.1. Endotoxin in Alzheimer’s Disease (AD)

Amyloid plaques and tau protein tangles are pathological hallmarks of AD, leading to cognitive and memory deficits and synaptic and neuronal loss [1]. According to many studies, endotoxin levels are increased in the brains of AD patients and can influence neurodegeneration via inflammation or direct impact on amyloid plaque deposition [131,132]. Using a rat model, Wang et al. [180] showed that endotoxemia causes neuroinflammation, expressed as increased levels of IL-1β, IL-6, and TNFα in the blood and brains of animals. The increase in inflammatory proteins’ iNOS and COX-2 expression in mice brains upon endotoxin stimulation was also reported by Lee et al. [181]. Zhan et al. [131] in AD brains evidenced with Western blot, DNA sequencing, and immunochemistry, the colocalization of endotoxin from *E. coli* K99 with Aβ1-40/42 in amyloid plaques and Aβ1-40/42 around vessels. Moreover, Ganz et al. [182] showed that, in transgenic mice carrying mutated genes associated with familial AD, the intravenous administration of endotoxin causes neuronal death in the cortex, unlike wild-type mice. In addition, neurodegeneration in the microglia-rich frontal cortex in AD mice directly affected endotoxin exposure without inducing Aβ deposition, the hallmark of AD and microgliosis. Endotoxemia was also associated with increased soluble Aβ and Aβ diffuse deposits in the mouse brain, which persisted throughout the observation period of 7–9 days. Moreover, endotoxin induced an increase in microglia density. Similarly, Kahn et al. [183] demonstrated that multiple injections of endotoxin resulted in increased Aβ1-42 deposits in the hippocampus and cognitive deficits in mice. According to Ahne et al. [92], the chronic induction of TLR4 by endotoxin can augment cytosolic levels of Ca^2+^, leading to apoptosis and impeding Aβ42 recognition and clearance, suggesting that gut dysbiosis-associated chronic asymptomatic endotoxemia may accelerate neurodegeneration in AD.

### 7.2. Endotoxin in Parkinson’s Disease (PD)

A progressive loss of midbrain dopaminergic neurons, which translates into motor disorders, is observed in PD, a common neurodegenerative disease affecting approximately 2% of people aged over 65 years in developed countries [184,185]. The critical pathological change in the CNS in these patients is the presence of intraneuronal aggregates of fibrillar α-synuclein, known as Lewy bodies and Lewy neurites. The role of endotoxin in developing PD, at least in a subset of patients, is confirmed by numerous studies that point to intestinal dysbiosis and endotoxemia as possible causes [1,186,187]. Peter et al. [188], in a retrospective cohort study, demonstrated that the incidence of PD among patients with inflammatory bowel disease (IBD) was 28% higher than the amount unaffected matched controls. In addition, anti-TNFα therapy for IBD correlated with a 78% reduction in the incidence rate of PD, confirming the role of systemic inflammation in the pathogenesis of the disease. Wijeyekonn et al. [189] evidenced increased endotoxin levels in the sera of PD patients with a high risk of early dementia, accompanied by changes in crucial surface markers of innate peripheral monocytes, i.e., α-synuclein and caspase-1. Forsyth et al. [190] revealed increased intestinal permeability compared to healthy controls in newly diagnosed PD patients—moreover, these patients’ level of leaky gut correlated with a-synuclein, oxidative stress, and Gram-negative bacteria. Next, de Waal et al. [191], using direct fluorescence in platelet-poor plasma of PD patients, showed increased endotoxin levels via monoclonal mouse anti-endotoxin antibodies associated with abnormal clotting. In rats’ primary mesencephalic cultures, Gayle et al. [192] demonstrated endotoxin cytotoxicity to dopamine neurons, directly caused by endotoxin-induced increased levels of TNFα and IL-1β. The endotoxin-triggered dopaminergic degeneration was partially abolished with specific anti-cytokine antibodies. According to a study by Bronstein et al. [193], endotoxin killed 70% of dopamine neurons in mixed neuronal–glial cultures in an in vitro model of Parkinson’s disease. In turn, Bodea et al. [194] demonstrated in vivo that a repeated systemic challenge of mice over four consecutive days with bacterial endotoxin maintained an elevated microglial inflammatory phenotype and induced the loss of dopaminergic neurons in the substantia nigra (SN). In addition, whole-brain transcriptome analysis revealed that 60 genes, mainly immune-related genes, e.g., complement components, Fc receptors, and MHC molecules, were selectively upregulated after repeated endotoxin challenge. Moreover, KEGG analysis showed that the complement cascade was the most vital common inflammatory pathway from both single and repeated endotoxin mice challenge, with complement C3 as an intermediate molecule essential in the loss of dopaminergic neurons triggered by systemic, repeated endotoxin application [194].

### 7.3. Endotoxin in Amyotrophic Lateral Sclerosis (ALS)

The direct impact of endotoxin on pathogenesis is also well documented in amyotrophic lateral sclerosis (ASL), which is an adult neurodegenerative fatal disease characterized by a progressive loss of function of motor neurons in the brain and spinal cord [174]. ALS patients show chronic, low-grade systemic inflammation with elevated IL-6, IL-1β, and TNFα in the blood, which correlates with the degree of clinical disability, disease progression, and inflammatory markers, i.e., C-reactive protein (CRP), endotoxin-binding protein (LBP), and secreted CD14 (sCD14) molecules [195,196]. According to the research by Keizman et al. [195], although the endotoxin level in the serum of ALS patients showed an increasing trend, it did not reach a significant difference with the endotoxin level in healthy people, in contrast to the LBP and sDC14 levels, which were significantly increased in ALS, especially in individuals with the rapidly progressive form of the disease. However, increased levels of endotoxin-binding proteins, i.e., LBP and sCD14, in these patients compared to healthy controls could be responsible for low endotoxin levels in the serum of ALS patients. In contrast, Zhang et al. [43] found significantly increased endotoxin levels in the blood of patients with sporadic ALS (sALS) compared to healthy controls, which correlated with the monocyte/macrophage activation level.

### 7.4. Endotoxin in Huntington’s Disease (HD)

Cognitive, motor, and psychiatric disorders are associated with rare, inherited HD diseases characterized by an autosomal mutation responsible for an increase in the number of CAG repeats in the huntingtin (HTT) gene. The mutation leads to an abnormally long polyglutamine expansion in the mutated huntingtin protein (mHTT) form that becomes neurotoxic [197,198]. In microglia, mHTT via the positive regulation of the NF-kB signaling pathway induces the release of several inflammatory markers, such as IL-1β, IL-6, TNFα, and IL-8, among others [199,200]. Interestingly, proinflammatory microglia activation is detectable in pre-symptomatic HD patients, suggesting that mHTT plays a vital role in the development of the disease [199]. On the other hand, Steinberg et al. [199] on mHD knock-in (Q140) and wild-type (Q7) mice models demonstrated that external triggers, e.g., endotoxin or proinflammatory signals, are required for microglia activation in HD. Still, a cell-autonomous dysfunction affecting HD microglia’s ability to acquire tolerance might contribute to establishing neuroinflammation in HD. Previously, Donley et al. [201] revealed that mHTT alters microglia immune responses depending on the nature of the inflammatory stimuli. Their study showed mutant huntingtin-expressing cells had higher basal NF-κB, which increased after IL-6 stimulation. In this context, it seems that, not necessarily endotoxin, but any infectious stimulant can induce peripheral inflammation, increasing the release of proinflammatory mediators activating brain microglia [202]. Furthermore, evidence of peripheral endotoxin-induced inflammation contributing to microglia activation was presented by Batista et al. [130], who showed that 12-month-old YAC128 transgenic mice (expressing the human mutant huntingtin protein) challenged for four months with peripherally injected low doses of endotoxin showed enhanced microglial alterations and vascular dysfunction in the BBB. The authors also observed that endotoxin exposure caused the increased nuclear localization of p65, an NF-κB subunit, in astrocytes and microglia in the cortex of transgenic mice, contributing to neuroinflammation [130]. Interestingly, attention has recently been drawn to intestinal dysbiosis and related disorders, such as weight loss, nutrient deficiency, and disturbances in gut structure, motility, and permeability, in patients with HD [203,204,205,206]. Gubert et al. [203] on a mice model demonstrated that dietary fiber interventions may have therapeutic potential in HD to delay clinical onset, suggesting that leaking gut and endotoxemia can influence HD progression indirectly via systemic inflammation.

## 8. Endotoxin-Induced Epigenetic Regulation of the Microglial Phenotype

The epigenetic regulation of gene expression by altering transcriptional activity without changing the DNA sequence includes DNA methylation, histone modifications, and noncoding RNAs [207]. Emerging evidence has highlighted that epigenetic mechanisms impact the expression and suppression of genes encoding biologically active proteins that control many cellular processes, leading to the appearance of desired or undesired features and functions. This can lead to the development of both desirable and undesirable features and functions. Neurodegeneration is a complex process in which numerous genetic and environmental factors interact, and epigenetic mechanisms are considered as a coupling factor. Neurodegeneration is a complex process influenced by genetic and environmental factors, with epigenetic mechanisms acting as a connecting factor. Recent evidence suggests that environmental factors, like diet and lifestyle, can change cellular functions, including gene expression. Therefore, it is clear that epigenetic mechanisms are highly significant in the development of neurodegenerative diseases. A growing body of evidence indicates that, at the genetic level, epigenetic modifications affect brain processes, such as memory, cognition, and motor functions. Moreover, these changes are often associated with microglial phenotypes, protecting or inducing neurodegenerative changes in the brain [208]. Microglia activation, associated with morphological, molecular, and functional remodeling in response to brain challenges (e.g., inflammation, protein misfolding, and aggregation, such as Aβ and tau in AD, α-synuclein in PD, and TAR DNA-binding protein 43 in ALS), is considered a primary factor contributing to the onset and progression of neurodegeneration [208,209,210]. Hence, factors favoring excessive microglial activation, including oxidative stress and systemic inflammation, can contribute to neurodegeneration [211]. Dunn et al. [212] have demonstrated that particular gene polymorphisms interact with exposure to environmental factors, such as cigarette smoking, pesticides, or coffee, to affect the risk for neurodegeneration in AD and PD differentially. Similarly, genetic variants can also influence the differential epigenetic regulation of innate immune responses to endotoxin. Recent data indicate that innate immune memory critically depends on epigenetic reprogramming, enhancing the immune cell’s capacity to respond appropriately to stimulation [213]. The potential impact of understanding how these epigenetic regulatory mechanisms become dysfunctional in cases of ME-associated chronic inflammation is immense, considering the growing number of obese individuals in developed countries. Further research in this area could pave the way for novel therapeutic interventions.

The following examples illustrate the potential role of endotoxin in inducing epigenetic changes in microglia.

### 8.1. DNA Methylation

One representative epigenetic modification profoundly affecting gene expression is DNA methylation, a process crucial for proper development, e.g., silencing retroviral elements, regulating tissue-specific gene expression, genomic imprinting, and X chromosome inactivation, and being involved in the pathomechanism of different diseases, including those associated with neurodegeneration [214,215]. In mammalian DNA, methylation mainly occurs on cytosines within CpG dinucleotides, and approximately 60% of human gene promoters contain clusters of CpGs referred to as CpG islands [216]. DNA methylation is catalyzed by enzymes from the DNA methyltransferase family (DNMTs), which transfer the methyl group from S-adenyl methionine (SAM) to the 5′ position of the pyrimidine ring of cytosine residues adjacent to guanines in the genome (CpG dinucleotides) to form 5-methylcytosine (5mC). DNA methylation changes the structure of chromatin, suppressing the binding of transcription factors and eventually silencing gene expression [214,217]. The 5mC further may undergo enzymatic oxidation to the oxidized form of 5-hydroxymethylcytosine (5hmC) and other oxidative derivates via TET (ten-eleven translocation) methylcytosine dioxygenases responsible for finely tuned methylation patterns [172]. While 5mC is associated with inhibiting gene expression, 5hmC has been associated with increased gene expression and is involved in cellular processes, such as differentiation, development, and aging. Accelerated DNA methylation, promoting heightened microglial activation, is a hallmark of aging, but is also found in neurodegenerative diseases such as PD, HD, and AD [215,217]. The direct capability of endotoxin to methylate the DNA of microglial cells has yet to be extensively studied. However, as a TLR4 agonist, endotoxin can impact microglia epigenetics by inducing inflammatory responses and cytokine production. It has been demonstrated that TLR4 expression increases in AD patients’ microglia and microglia surrounding amyloid plaques [218]. The activation of TLR4 by PAMPs and DAMPs leads to the translocation of NFĸB to the cell nucleus and the production of pro-inflammatory factors, such as cytokines, chemokines, and nitric oxide. IL-10, IL-6, IL-8, and TNFα levels are substantially associated with DNA methylation patterns in peripheral blood mononuclear cells [219]. Stimulating peripheral blood mononuclear cells with endotoxin decreases mitochondrial DNA methylation and strongly correlates with IL-6 and IL-10 expression. However, whether these cytokines, while present in circulation or produced locally in microglia, can influence DNA methylation patterns in microglial cells needs to be determined [166]. On the other hand, a direct effect of endotoxin on DNA methylation in microglia has been shown in adult (4–6-month-old) C57BL/6 mice. The stimulation of murine primary microglia with endotoxin decreased the methylation of the IL-1β promoter, contributing to an increase in IL-1β gene expression and intracellular IL-1β production, which suggests that ME, especially in the elderly, can disturb the DNA methylation of IL-1β, leading to microglial dysfunction [217].

### 8.2. MicroRNAs

MicroRNAs (miRNAs) are small noncoding RNAs that regulate gene expression by recognizing cognate sequences and interfering with transcriptional, translational, or epigenetic processes. They are short RNA molecules ranging from 19 to 22 nucleotides in length that play a crucial role in posttranscriptional gene silencing by targeting multiple genes located within the same cellular signaling pathway [220]. MiRNAs affect many neurobiological processes, such as cell growth and proliferation, apoptosis, tissue differentiation, and embryonic development, and have been predicted to regulate up to 90% of human genes [221]. Increasing evidence indicates that miRNAs are essential regulators mediating microglial activation, polarization, and autophagy, and, thus, can affect neuroinflammation [180,222]. miRNAs can regulate microglia-mediated neuroinflammation by targeting relevant cellular signaling pathways, and notable examples involved in microglial activation and most strongly associated with proinflammatory pathways include miR-155, miR-146a, miR-124, and miR689 [180,223]. On the contrary, MiR-711 and miR-145 strongly mediate anti-inflammatory pathways in microglia [180]. The effect of endotoxin on miRNA regulation in microglia is well evidenced, although it primarily concerns miRNA regulation in the polarization of microglia and macrophages. Several miRNAs regulate immune cell signaling upon endotoxin challenge [224]. The NFĸB-dependent endotoxin-induced upregulation of miR146a in human monocytes, critical for endotoxin-induced immune tolerance, has been shown [225,226]. In turn, miR-155 is upregulated in microglial cells, which switch from resting to the primed state upon treatment with endotoxin. miR-155 targets and suppresses several downstream TLR4 mediators, regulating inflammation and endotoxin tolerance in BV2 microglial cells [227]. Similarly, N9 microglial cells challenged with endotoxin enter the primed state, confirmed by enhanced phagocytosis, NLRP3-inflammasome activation, upregulated miR-155 and miR-146a, and downregulated miR-124. Interestingly, the process is replicated in N9 cell-derived exosomes, likely contributing to the regulation of the inflammatory response of recipient cells and dissemination processes [227,228].

### 8.3. Histone Modifications

Histone proteins are spools around which DNA winds, creating a structural nucleosome unit. Each unit comprises two copies of the canonical histones H3, H4, H2A, H2B6, and histone linker H1/H5 [229]. Histones may be subject to modifications, such as acetylation, methylation, ubiquitination, phosphorylation, sumoylation, adenylation, and glycosylation, which play an essential role in the onset and progression of neurodegeneration [230]. Histone acetylation, supervised by histone acetyltransferases (HATs), reduces the affinity between DNA strands and histones, causing the loosening of the chromatin structure and facilitating the binding of transcription factors to DNA, eventually activating gene transcription. In contrast, histone deacetylation, carried out by histone deacetylases (HDACs), relies on removing acetyl groups from histones, tightening chromatin structure, and inhibiting gene transcription [208]. Both these processes provide synaptic plasticity, cognitive functions, and memory processing [231,232]. Endotoxin-induced histone acetylation in murine microglial cell line BV2 provides feedback and attenuates cytokine-induced inflammatory responses [233]. Moreover, endotoxin challenge induced the expression of the gene suppressor of cytokine signaling 3 (SOCS-3) via the histone H3 and H4 acetylation of SOCS-3, promoting the rapid activation of MAPK signaling pathways (ERK1/2, p38, and JNK), the induction of IL-10, and the subsequent activation of STAT-3 [233,234].

## 9. Conclusions

Endotoxin, as a TLR4 agonist, is an inflammatory stimulator most frequently used in research models of inflammation in vitro and in vivo. Differences in the immunogenicity of endotoxin, depending on the structure of lipid A, and the imperfections of available tests for detecting endotoxin in the blood of tested individuals, which do not differentiate between immunogenic and non-immunogenic endotoxin, lead to controversial research results on the impact of ME on the brain. Additionally, fluctuations in endotoxin levels in human blood, depending on the type and composition of a meal and the long duration of endotoxemia, complicate the evaluation of the actual role of ME in neurodegenerative processes. However, growing evidence strongly suggests that epigenetic changes in microglia that occur with aging make the human body more sensitive to even very low endotoxin levels, explaining the increase in the incidence of neurodegenerative diseases in the elderly. In the future, detailed knowledge of the epigenetic mechanisms related to neurodegeneration, including the role of miRNAs, may allow the silencing of excessively expressed miRNAs using artificial antagonists, preventing the development of these diseases. However, in the meantime, the available data indicate that neurodegenerative processes develop over many years, and the influence of external factors, such as diet or lifestyle, on these processes is undeniable. Therefore, until we obtain effective methods for treating neurodegenerative diseases, it is essential to promote professional knowledge on this subject.

## Figures and Tables

**Figure 1 ijms-25-07006-f001:**
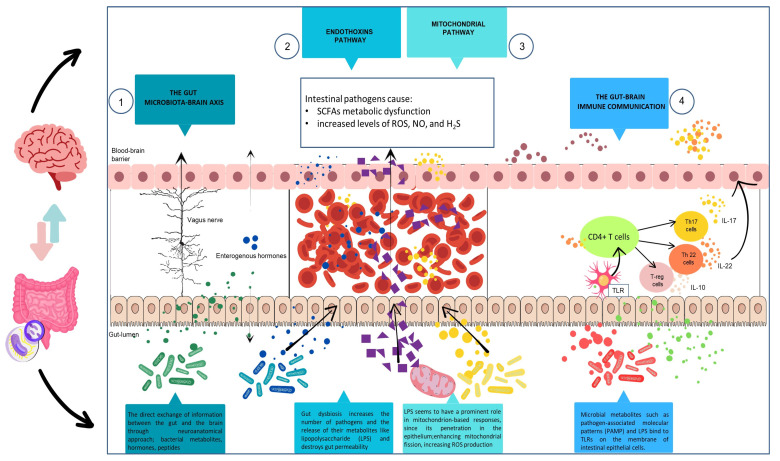
Schematic diagram of two-way communication connecting intestinal microflora with the brain. The transmission between the dysbiotic gut and the central nervous system occurs through several pathways: (1) the neuroanatomical pathways (vagus nerve, VN), where vagal afferent fibers originate from terminals in the intestinal wall, but do not have direct access to the intestinal microflora. The vagal fibers cannot penetrate the intestinal epithelium unless the intestinal epithelium’s integrity is destroyed, which creates the possibility of microorganism invasion. When the intestinal epithelial barrier is disrupted, the gut bacteria and their metabolites can make direct contact with the VN. (2) The intestine contains numerous bacteria that produce endotoxins or can convert dietary components into several metabolites, such as SCFAs. These metabolites regulate homeostasis, maintain BBB integrity, and influence brain function. Microbial dysbiosis leads to a proinflammatory milieu and systemic endotoxemia, contributing to the development of neurodegenerative diseases and metabolic disorders. The enteric vascular barrier (GVB) prevents bacteria from entering the bloodstream. After the destruction of the GVB, the bacteria and their toxic metabolites enter the bloodstream, causing oxidative/nitrative stress and inflammatory reactions. (3) Proinflammatory factors involved in local intestinal inflammation may reach the brain to induce mitochondrial dysfunction in microglia. (4) TLR receptors recognize ligands from commensal or pathogenic bacteria to maintain tolerance or trigger immune response, respectively. TLR4 involvement induces several intracellular signaling cascades, producing cytokines and chemokines essential for maintaining gut homeostasis and infection control. CD4+ T cells produce proinflammatory cytokines (such as interleukin-10 (IL-10) and interleukin-17 (IL-17)), which may enter the peripheral circulation driving systemic inflammation. Interleukin-22 (IL-22) is a cytokine with many protective qualities, but requires constant regulation to prevent overproduction in inflammatory settings. IL-22 is considered protective at barrier sites exposed to external stimuli in an acute setting. However, chronic inflammation can result in the dysregulation of IL-22 signaling, promoting overt tissue damage.

**Figure 2 ijms-25-07006-f002:**
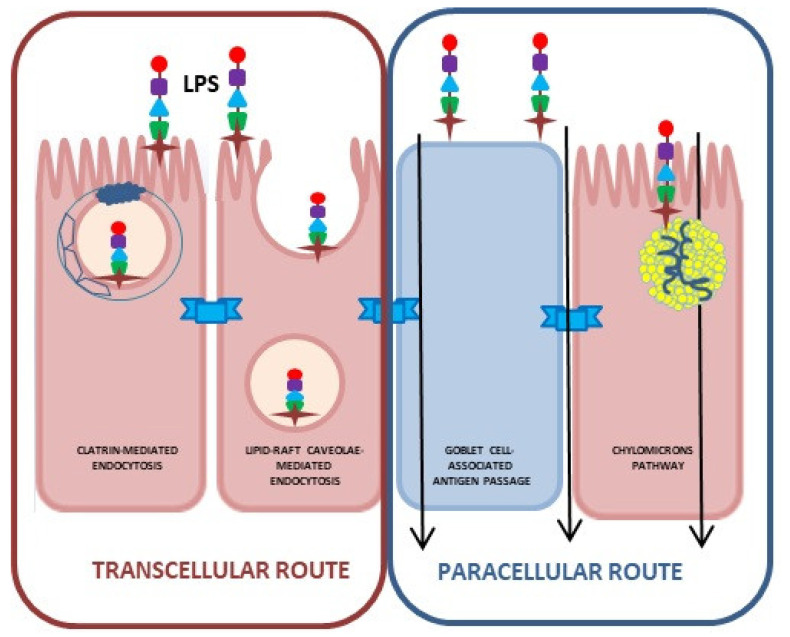
Transfer of endotoxin from the gut to circulation via transcellular and paracellular routes. Endotoxin enters the circulation through two routes: (1) paracellular transport through epithelial cell tight junctions and (2) transcellular transport through lipid raft membrane domains involving receptor-mediated endocytosis. Paracellular transport of endotoxins occurs through the dissociation of tight junction protein complexes. Transcellular transport through specialized membrane areas rich in glycolipids, sphingolipids, cholesterol, and saturated fatty acids is the result of the participation of a raft of endotoxin-related signaling proteins. This results in endotoxin signaling and endocytosis.

**Figure 3 ijms-25-07006-f003:**
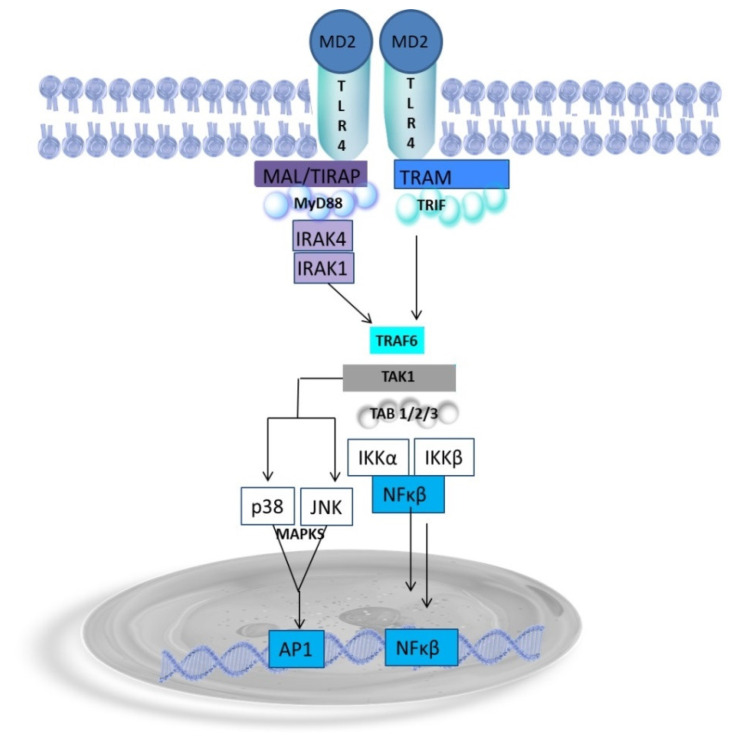
The MyD88-independent signaling pathway. Recognition of LPS activates MyD88-dependent and TRIF-dependent pathways. Recruitment of MyD88 by TIRAP initiates the interaction of IRAKs and TRAF6, activating transcription factors. AP1, activated protein 1; IRAK, interleukin receptor-associated kinase; IKK, inhibitor of k B kinase; JNK, c-Jun N-terminal kinase; MAPK, mitogen-activated protein kinase; MyD88, myeloid differentiation primary response protein 88; NF-kB, nuclear factor-kB; p38, protein 38; TAK1, transforming growth factor β-activated kinase 1; TRIF-related adaptor molecule; TIRAP, TIR domain-containing adaptor protein.

**Figure 4 ijms-25-07006-f004:**
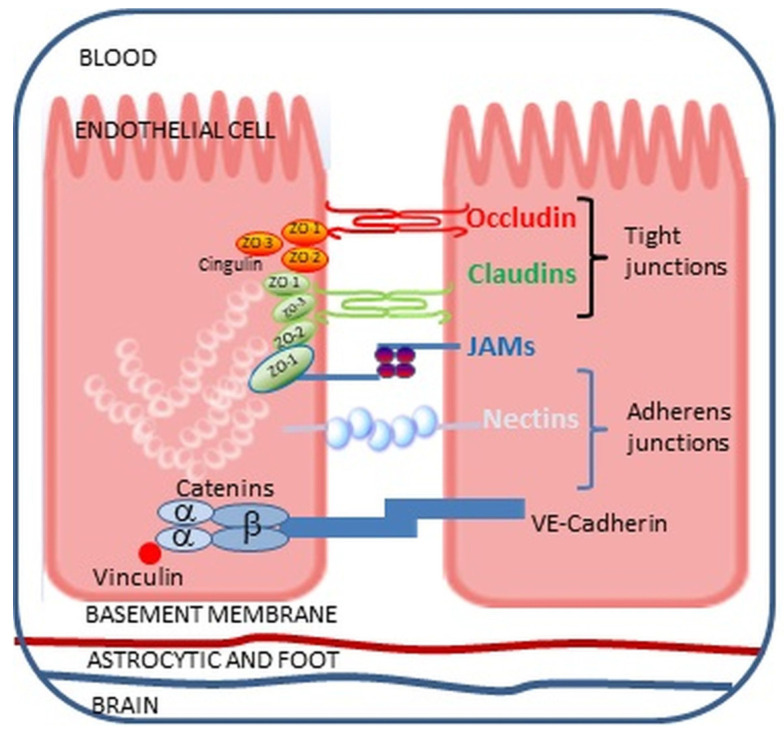
Transport of endotoxin through the blood–brain barrier. The tight junction proteins include claudin, occludin, and zonula occludins (ZO-1,2,3). Claudin and occludin are both transmembrane proteins, while zonula occludens are intracellular proteins. The cytoplasmic catenins form a complex with Ve-cadherin (JAMs) junctional adhesion molecules; the cingulin cytosolic protein can bind to and bundle actin filaments and interact with myosin II and several TJ proteins, including ZO-1, ZO-2, and ZO-3. Catenins are a family of polypeptides that bind to the conserved cytoplasmic tail of cadherins and are required for cadherin function.

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
