# Peer review of "Metabolic Endotoxemia: From the Gut to Neurodegeneration"

_ijms, 2024, doi:10.3390/ijms25137006_

Round 1

Reviewer 1 Report

Comments and Suggestions for Authors

The authors in this review described evidence from literature on the pathway trough which endotoxin crosses the gut towards the central nervous system, contributing there in potential development of neurodegenerative diseases via impact on microglia.

The topic is interesting but this work presents a major problem relating to clarity and structure.

In particular,

1) the abstract section is too short, and it should be a little more detailed. Check for the numbers of words for this section;

2) the authors should start their work with an introduction section where they should introduce the major topic, what is known from the literature, what it is still needed to know, the major aim of the work and how they want to structure the review. This is very important, in order to be clear on the main objectives of the work;

3) the authors use endotoxin and LPS as acronyms, but to avoid confusion, it would be better to use one or the other;

4) the authors should number the subsequent paragraphs after the introduction in a logical order. For instance, whether the paragraph “Factors promoting ME development” is numbered 2, “Diet” should be named 2.1 ..;

5) the different paragraphs should be connected among them, and it is needed that there is a sentence or more sentences that bind one from the other, otherwise it is very difficult following the whole text.

6) the Figure 1 represents the transcellular and paracellular routes, whereas the Figure 2 a detail of paracellular transport. I think this is redundant and not too much informative for the major aim of the whole work. Moreover, the legends should be more detailed.

For instance, the authors should report a Figure that describes all details on the different steps of the endotoxin from gut, blood, pro-inflammatory response, blood-brain barrier and potential neurodegeneration …;

7) the paragraph “Can LPS cross the blood brain barrier?” is too long and very difficult to follow; as well as “the impact of endotoxin on the blood brain barrier”;

8) the paragraph “microglia” shows the same problem; it can be removed because the authors describe the microglia functions in physiological condition which in this context are misleading.

Comments on the Quality of English Language

Enghish is fine. 

Author Response

1) the abstract section is too short, and it should be a little more detailed. Check for the number of words for this section;

Answer: The abstract has been expanded and detailed, and now contains 189 words.

2) the authors should start their work with an introduction section where they should introduce the major topic, what is known from the literature, what it is still needed to know, the major aim of the work and how they want to structure the review. This is very important, in order to be clear on the main objectives of the work;

Answer: The introduction section has been added.

3) the authors use endotoxin and LPS as acronyms, but to avoid confusion, it would be better to use one or the other;

Answer: Following the reviewer's suggestion, the term endotoxin replaced the abbreviation LPS throughout the manuscript, except for the chapter on metabolic endotoxemia, where endotoxin was described for the first time as lipopolysaccharide.

4) the authors should number the subsequent paragraphs after the introduction in a logical order. For instance, whether the paragraph “Factors promoting ME development” is numbered 2, “Diet” should be named 2.1 ..;

Answer: The subsequent paragraphs have been numbered in the revised version of the manuscript.

5) the different paragraphs should be connected among them, and it is needed that there is a sentence or more sentences that bind one from the other, otherwise it is very difficult following the whole text.

Answer: According to the reviewer's suggestion, we have tried to combine individual subsections wherever possible.

6) Figure 1 represents the transcellular and paracellular routes, whereas Figure 2 is a detail of paracellular transport. I think this is redundant and not too much informative for the major aim of the whole work. Moreover, the legends should be more detailed.

For instance, the authors should report a Figure that describes all details on the different steps of the endotoxin from gut, blood, pro-inflammatory response, blood-brain barrier and potential neurodegeneration …;

Answer: Both figures presenting the transport of endotoxin from the gut to the brain have been removed in the revised version of our manuscript, and a new figure pointing to the pathway of endotoxin from the intestines to the CNS has been added.

7) the paragraph “Can LPS cross the blood brain barrier?” is too long and very difficult to follow; as well as “the impact of endotoxin on the blood brain barrier”;

Answer: As suggested by the reviewer, both paragraphs have been shortened.

8) the paragraph “microglia” shows the same problem; it can be removed because the authors describe the microglia functions in physiological condition which in this context are misleading.

Answer: The paragraph Microglia has been removed in a revised version of our manuscript.

Reviewer 2 Report

Comments and Suggestions for Authors

It is recommended that the authors complete the abstract with a summary of the key points and findings of the review. This would provide a concise overview of the document's content, allowing readers to quickly grasp the essence of the review.

The document discusses transcellular and paracellular routes but could benefit from an expanded discussion on the specific molecular mechanisms and pathways involved in the transfer of endotoxins from the gut to the bloodstream.

The review notes the varied responses of microglia to different levels of lipopolysaccharides. Delving deeper into the signaling pathways and molecular mechanisms that drive these responses would provide more detailed and valuable insights.

The section on the epigenetic regulation of microglia is very interesting but lacks comprehensive coverage. Including a more detailed analysis of specific epigenetic modifications and their effects on microglial function and neuroinflammation would complete it.

Although the document references several animal studies, it should also focus on human studies and clinical trials that would offer a clearer understanding of how these findings may translate to clinical applications.

Lastly, the authors should check that all citations are formatted consistently in accordance with the journal's guidelines.

Comments on the Quality of English Language

The English quality of the manuscript requires minor editing

Author Response

It is recommended that the authors complete the abstract with a summary of the key points and findings of the review. This would provide a concise overview of the document's content, allowing readers to quickly grasp the essence of the review.

Answer:   According to the reviewer's suggestion the abstract has been abstract has been expanded based on the manuscript content.

The document discusses transcellular and paracellular routes but could benefit from an expanded discussion on the specific molecular mechanisms and pathways involved in the transfer of endotoxins from the gut to the bloodstream.

The review notes the varied responses of microglia to different levels of lipopolysaccharides. Delving deeper into the signaling pathways and molecular mechanisms that drive these responses would provide more detailed and valuable insights.

The section on the epigenetic regulation of microglia is very interesting but lacks comprehensive coverage. Including a more detailed analysis of specific epigenetic modifications and their effects on microglial function and neuroinflammation would complete it.

Although the document references several animal studies, it should also focus on human studies and clinical trials that would offer a clearer understanding of how these findings may translate to clinical applications.

Answer: Following the reviewer's instructions, we made some corrections, but not all of them due to significant time constraints during this period. We hope that the corrections made will be sufficient to consider our manuscript worthy of publication in the International Journal of Molecular Sciences.

Lastly, the authors should check that all citations are formatted consistently in accordance with the journal's guidelines.

 Answer: All citations have been checked.

Round 2

Reviewer 1 Report

Comments and Suggestions for Authors

The authors responded to most of my comments.

Indeed, I still have some points to clarify:

1. after the aim of the work, it would be useful to introduce how the authors structured the review.

2. What is the figure that summarizes the endotoxin pathways that the authors said they added? Figure 2?

3.3. Are figure 1 and figure 3 indicated in the text?

4. In the text, figure 3 is shown before figure 2. 

5. The conclusions could also refer to what the clinical impact of the topic covered is, especially in terms of identifying new pharmacological targets.

Author Response

Dear Reviewer,

Thank you for your valuable feedback on our manuscript. We appreciate your detailed review and have addressed your comments as follows:

Reviewer’s Comment: "After the aim of the work, it would be useful to introduce how the authors structured the review."

Response: We have added a brief section after the aim of the work outlining the structure of the review. This section now provides a clear roadmap for the reader, detailing the organization of the content and the flow of the arguments presented.

Reviewer’s Comment: "What is the figure that summarizes the endotoxin pathways that the authors said they added? Figure 2?"

Response: We apologize for any confusion. The figure summarizing the endotoxin pathways is indeed Figure 2. We have revised the text to make this clearer.

 Reviewer’s Comment: "Are figure 1 and figure 3 indicated in the text?"

Response: We have carefully reviewed the manuscript and ensured that both Figure 1 and Figure 3 are now properly indicated in the text at appropriate points.

 Reviewer’s Comment: "In the text, figure 3 is shown before figure 2."

Response: We have corrected the sequence of figures in the text so that Figure 2 is introduced before Figure 3. This change aligns the order of figures with their mention in the manuscript.

Reviewer’s Comment: "The conclusions could also refer to what the clinical impact of the topic covered is, especially in terms of identifying new pharmacological targets."

Response: We have expanded the conclusion section to discuss the clinical implications of our findings. Specifically, we have included a discussion on potential new pharmacological targets identified through our review, emphasizing their significance for future clinical applications.

We hope these revisions adequately address your comments and improve the clarity and impact of our manuscript. Thank you once again for your insightful feedback.

Reviewer 2 Report

Comments and Suggestions for Authors

The article was competed according to my inquiries, so I consider it proper to be accepted

Author Response

Dear Reviewer,

Thank you for your review and feedback on our manuscript. We are pleased to hear that the revisions have addressed your inquiries and that you consider the article suitable for acceptance.